# Evaluation and Comparison of Lattice-Based Cryptosystems for a Secure Quantum Computing Era

Maria E. Sabani [1,*,†] , Ilias K. Savvas [1,†] , Dimitrios Poulakis [2] , Georgia Garani [1] and Georgios C. Makris [1]

1 Department of Digital Systems, University of Thessaly, Geopolis Campus, Larissa-Trikala Ring-Road, 415 00 Larissa, Greece; isavvas@uth.gr (I.K.S.); garani@uth.gr (G.G.); makris@uth.gr (G.C.M.)
2 Department of Mathematics, Aristotle University of Thessaloniki, 541 24 Thessaloniki, Greece; poulakis@math.auth.gr
* Correspondence: masampani@uth.gr
† These authors contributed equally to this work.

**Abstract:** The rapid development of quantum computing devices promises powerful machines with the potential to confront a variety of problems that conventional computers cannot. Therefore, quantum computers generate new threats at unprecedented speed and scale and specifically pose an enormous threat to encryption. Lattice-based cryptography is regarded as the rival to a quantum computer attack and the future of post-quantum cryptography. So, cryptographic protocols based on lattices have a variety of benefits, such as security, efficiency, lower energy consumption, and speed. In this work, we study the most well-known lattice-based cryptosystems while a systematic evaluation and comparison is also presented.

**Keywords:** quantum computing; lattice-based cryptosystems; post-quantum cryptography

## 1. Introduction

Quantum computing constitutes a critical issue as the impact of its advent and development will be present in every cell of our technology and therefore, our life. Quantum computational systems use the qubit (QUantum BIT) instead of the typical bit, which has a unique property; it can be in basic states $|0\rangle$, $|1\rangle$ but it can also be in a state that is a linear combination of these two states, such that $a|0\rangle + b|1\rangle, a, b \in \mathbb{C}, \wedge a^2 + b^2 = 1$ [1]. This is an algebraic-mathematical expression of quantum superposition which claims that two quantum states can be added and their sum can also be a valid quantum state [2]. Regardless of superposition, quantum computers' power and capability are based on quantum physics and specifically on the phenomenon of quantum entanglement and the no-cloning system. The odd phenomenon of quantum entanglement states that there are particles that are generated, interact, and are connected, regardless of the distance or the obstacles that separate them [3]. This fundamental law of quantum physics allows us to know or to measure the state of one particle if we know or measure the other particles.

Programmable quantum devices are capable of solving and overcoming problems that typical computers cannot solve in logical time. A quantum computer can perform operations with enormous speed, and in the flash of an eye, can process and store an extensive amount of information. This huge computational power which makes quantum computers superior to classical computers was described in 2012 by John Preskill with the term quantum supremacy [4]. Quantum mechanics provides us a fascinating theorem, the no-cloning theorem. As an evolution of the no-go theorem by James Park, the no-cloning theorem was proposed, a fundamental theorem of quantum physics and quantum cryptography. According to this theorem, the independent and identical replication of any unknown quantum state is impossible [2].

Cryptography is one of the oldest sciences and was developed out of the human necessity for secure communication [5]. Cryptographic protocols and algorithms are

based on complex mathematics and cryptosystems appear in every electronic transaction and communication in our everyday life. The security, efficiency, and speed of these cryptographic methods and schemes are a main issue of interest and study. Contemporary cryptosystems are considered to be vulnerable to a quantum computer attack. In 1994, the American mathematician and cryptography professor Peter Shor presented an algorithm [6], which dumbfounded scientists. Shor in his work argued that with the implementation of the proposed algorithm in a quantum device, there would be no more security in current computational systems. This was a real revolution for the science of computing and a great motivator for the design and construction of quantum computational devices. The science that studies and develops cryptographic algorithms resistant to attacks by quantum computers is well known as post-quantum cryptography [7]. By bringing up to date mathematically based algorithms and standards, post-quantum cryptography examines and studies how to prepare the world for the era of quantum computing. [8,9].

Lattice-based cryptographic protocols attract the interest of researchers for a number of reasons. Firstly, the algorithms that are applied to lattice-based protocols are simple and efficient. Additionally, they have proven to be secure protocols and create a multitude of applications.

In this review, we examine the cryptographic schemes that are developed for a quantum computer. The following research questions are answered:

- How much is the science of Cryptography affected by quantum computers ?
- Which cryptosystems are efficient and secure for the quantum era?
- Which are the most known lattice-based cryptographic schemes and how do they function?
- How can we evaluate NTRU, LWE, and GGH cryptosystem?
- What are their strengths and weaknesses?

The rest of the paper is organized as follows. In Section 2, we present changes and challenges due to quantum devices in cryptography and in Section 3, cryptographic schemes in the quantum era are described. In Section 4, we present some basic issues of lattice theory. In Sections 5 and 6, we present the lattice-based cryptographic schemes NTRU, LWE, and GGH correspondingly, while a discrete implementation of them is given. In addition, the GGH cryptosystem is described in Section 7. Results and comparisons are given in Section 8, while some future work directions are presented in Section 9. Finally, Section 10 concludes this work.

## 2. The Evolution of Quantum Computing in Cryptography

Cryptography is an indispensable tool for protecting information in computer systems, and difficult mathematical problems such as the discrete logarithm problem and the factorization of large prime numbers are the basis of current cryptographic protocols. We can divide the cryptographic protocols into two broad categories: symmetric cryptosystems and asymmetric (public key cryptosystems) cryptosystems [5].

The same key for both encryption and decryption is being used in symmetric cryptosystems, and despite their speed and their easy implementation, they have certain disadvantages. One main issue of this type of cryptosystem is the secret key distribution between two parties that want to communicate safely. Another drawback of symmetric cryptographic schemes is that the private keys which are being used must be changed frequently in order not to be known by a fraudulent user. If we can ensure the existence of an efficient method to generate and exchange keys, symmetric encryption and decryption methods are considered to be secure [10,11].

Asymmetric cryptographic schemes use a pair of keys, private and public keys, for encryption and decryption. This type of cryptosystem relies on mathematical problems that are characterized as hard to solve [12]. Some of the most widely known and implemented public key cryptosystems are RSA [13], the Diffie–Helman protocol, ECDSA, and others. Since the early 1990s, all these cryptographic schemes were believed to be effective and secure but Shor's algorithm changed things.

Peter Shor proved with his algorithm that a quantum computer could quickly and easily compute the period of a periodic function in polynomial time [14]. Since 1994, when Shor's protocol was presented, has been a great amount of study, analysis, and implementation of the algorithm both in classical and quantum computing devices. Shor's method solves both the discrete logarithm problem and the factorization problem that are the basis of the current cryptographic schemes and therefore, the public key cryptosystems are insecure and vulnerable to a quantum attack [6].

### 2.1. Quantum Cryptography

In 1982, for the first time the term "Quantum Cryptography" was recommended but the idea of quantum information appeared for the first time in the decade of the 1970s, from Stephen Wiesner and his work about quantum money [15]. The science of quantum cryptography uses the fundamental laws of quantum physics to securely transfer or store data. In general, in quantum cryptography, the transmission and the encryption procedure is performed with the aid of quantum mechanics [16]. Quantum cryptography exploits the fundamental laws of quantum mechanics such as superposition and quantum entanglement, and constructs cryptographic protocols in a more advanced and efficient way.

A basic problem in classical cryptographic schemes is the key generation and exchange, as this process is endangered and unsafe when it takes place in an insecure environment [17]. When two different parties want to communicate and transfer data, they exchange information (i.e., key, message) and this procedure occurs in a public channel, so their communication could be vulnerable to an attack by a third party [18]. The most fascinating and also the most useful discovery and a widely used method of quantum cryptography is quantum key distribution.

### 2.2. Quantum Key Distribution

Quantum key distribution (QKD) utilizes the laws of quantum physics in the creation of a secret key through a quantum channel. With the principles of quantum physics, in QKD a secret key is generated and a secure communication between two (or more parties) is established. The inherent randomness of the quantum states and the results accrue from their measurements and they have as a result total randomness in the generation of the key. Quantum mechanics solves the problem of key distribution—the main challenge in cryptographic schemes—with the aid of quantum superposition, quantum entanglement, and the Uncertainty Principle of Heisenberg. Heisenberg's Principle argues that two quantum states cannot be measured simultaneously [3]. This principle has as a consequence the detection of someone who tries to eavesdrop on the communication between two parties. If a fraudulent user tries to change the quantum system, he will be detected and the users abort the protocol.

Let us suppose that we have two parties that want to communicate and use a quantum key distribution protocol to generate a secret key. A quantum key distribution scheme has two phases and for its implementation the existence of a classical and a quantum channel is necessary. In the quantum channel, the private key is generated and reproduced and in the classical channel, the communication of the two parties takes place. Into the quantum channel are sent polarized photons and each one of the photons has a random quantum state [17]. Both the two parties have in their possession a device that collects and measures the polarization of these photons. Due to Heisenberg's principle, the measurement of the polarized photons can reveal a possible eavesdropper as in his effort to elicit information, the state of the quantum system changes and the fraudulent user is detected [19].

The BB84 protocol, named after its creators and the year it was published, was the first quantum key distribution protocol and it was proposed in 1984 by Charles Bennett and Gilles Brassard [20]. BB84 is the most studied, analyzed, and implemented QKD protocol; since then, various QKD protocols have been proposed. B92 and SARG04, which are known as variants of BB84 and E91 that exploit the phenomenon of quantum entanglement, are a few of the widely known quantum key distribution protocols [1]. All these QKD protocol

are in theory well designed and structured and are proved to be secure, but in practice, there are imperfections in their implementation. Loopholes, such as poorly constructed detectors or defective optical fibers, and general imperfections in devices and the practical QKD system make the QKD protocols vulnerable to attacks. By exploiting these weaknesses of the system, one can perform certain types of attacks and this is the basic issue of research and study, QKD security.

Significant progress has been made in the implementation of the quantum phase of communication and the development of quantum systems. Entanglement dynamics in CV quantum channels for both common and independent reservoirs have received a lot of attention recently [21]. As the security of QKD is the main goal, interesting experiments have shown that non-Markovian features can be used to improve security and/or locate an eavesdropper along the transmission line and determine their location [22]. Additionally, the entanglement dynamics have been studied and recent experiments have shown that photonic band gap media are promising to acquire non-Markovian behaviour and that materials with a photonic bandgap may be able to transmit entanglement reliably over long distances [23]. Moreover, the study of the phase modulation of coherent states in channels where the quantum communication phase takes place has turned into a subject of interest. Very interesting and useful studies and experiments have proven that phase diffusion is the most damaging kind of noise in a phase modulation scheme, where the information is encoded in the phase of a quantum seed signal [24]. Additionally, time-independent Markovian noise, specifically when the seed state is coherent, has been shown to be detrimental to information transfer and may compromise the channel's overall performance [25,26]. The environment's spectral structure, on the other hand, may lead to non-Markovian damping or diffusion channels in quantum optical communications [22,26]. It has also demonstrated that phase channels better preserve the transfer of information above a threshold on the loss and phase noise parameters, which is compared to the lossy coherent states amplitude-based scheme. So, in the presence of time-correlated noise, which results in dynamical non-Markovian phase diffusion, the interaction between the use of NLA and the memory effects results in a pronounced rise in performance [24].

## 3. Cryptographic Schemes in Quantum Era

The advances in computer processing power and the evolution of quantum computers seem for many people to be a threat in the distant future. On the other hand, researchers and security technologists are anxious about the capabilities of a quantum computational device to threaten the security of contemporary cryptographic algorithms. Shor's algorithm consists of two parts, a classical part and a quantum part, and with the aid of a quantum routine could break modern cryptographic schemes, such as RSA and the Diffie–Hellman cryptosystem [27]. The factorization problem and the discrete logarithm problem are the fundamental basis for modern cryptographic schemes and serve as the foundation for these kinds of cryptosystems.

From that moment and after, it has been widely known in the scientific and technological community that with the arrival of a sufficiently large quantum computer, there is no more security in our encryption schemes. Therefore, post-quantum data encryption protocols are a basic topic of research and work, with the main goal being to construct cryptosystems resistant to quantum computers' attacks [7,8]. Subsequently, we present certain cryptographic schemes that have been developed and that are secure under an attack of a quantum computer.

### 3.1. Code-Based Cryptosystems

Coding Theory is an important scientific field which studies and analyzes linear codes that are being used for digital communication. The main subject of research in coding theory is finding a secure and efficient data transmission method. In the process of data transmission, data are often lost due to errors owing to noise, interference, or other reasons, and the main subject of study of coding theory is to minimize this data loss [28]. When

two discrete parties want to communicate and transfer data, they add extra information to each message which is transferred to enable the message to be decoded despite the existing errors.

Code-based cryptographic schemes are based on the theory of error-correcting codes and are considered to be prominent for the quantum computing era. These cryptosystems are considered to be reliable and their hardness relies on hard problems of coding theory, such as syndrome decoding (SN) and learning parity with noise (LPN).

The first code-based cryptosystem was proposed by Robert McEliece in 1978. It was based on the difficulty of decoding random linear codes, a problem which is considered to be NP-hard [29]. The main idea of McEliece is to use an error-correcting code, for which a decoding algorithm is known and which is capable to correct up to *t* errors to generate the secret key. The public key is constructed by the private key, covering up the selected code as a general linear code. The sender creates a codeword using the public key that is disturbed up to *t* errors. The receiver performs error correction and efficient decoding of the codeword and decrypts the message.

McEliece's cryptosystem and the Niederreiter cryptosystem that was proposed by Harald Niederreiter in 1986 [30] can be suitable and efficient for encryption, hashing, and signature generation. The McEliece cryptosystem has a basic disadvantage, which is the large size of the keys and ciphertexts. In modern variants of the McEliece cryptosystem, there has been an effort to reduce the size of the keys. However, these types of cryptographic schemes are considered to withstand attacks by quantum computers and this makes them prominent for post-quantum cryptography.

### 3.2. Hash-Based Cryptosystems

Hash-based cryptographic schemes in general generate digital signatures and rely on cryptographic hash functions' security, such as SHA-3. In 1979, Ralph Merkle proposed an asymmetric signature scheme based on one-time signature (OTS) and the Merkle signature scheme is considered to be the simplest and the most widely known hash-based cryptosystem [31]. This digital signature cryptographic scheme converts a weak signature with the aid of a hash function to a strong one.

The Merkle signature scheme is a practical development of Leslie Lamport's idea of OTS that turn it into a many-times signature scheme, a signature process that could be used multiple times. The generated signatures are based on hash functions and their security is guaranteed even against quantum attacks.

Many of the reliable signature schemes based on hash functions have the drawback that the person who signs must keep record of the precise number of messages that have been signed before, and any error in this record will create a gap in their security [32]. Another disadvantage of these schemes is that a certain number of digital signatures can be generated and if this number increases indefinitely, then the size of the digital signatures is exceptionally big. However, hash-based algorithms for digital signatures are regarded as safe and strong against a quantum attack and can be used for post-quantum cryptography.

### 3.3. Multivariate Cryptosystems

In 1988, T. Matsumoto and H. Imai [33] presented a cryptographic scheme which relied on two-degree multivariate polynomials over a finite field for encryption and for signature verification. In 1996, J. Patarin [34] implemented a cryptosystem, the security of which relied on the fact that multivariate polynomial systems in finite fields are difficult to solve.

The multivariate quadratic polynomial problem states that given *m* quadratic polynomials $f_1, \ldots, f_m$ in *n* variables $x_1, \ldots, x_n$ with their coefficients to be chosen from a field $\mathbb{F}$, it is requested to find a solution $z \in \mathbb{F}^n$ such that $f_i(z) = 0$, for $i \in [m]$. The choice of the parameters make the cryptosystem reliable and safe against attacks, so this problem is considered to be NP-hard.

These type of cryptographic schemes are believed to be efficient and fast with high-speed computation processes and suitable for implementation on smaller devices. The

need for new, stronger cryptosystems with the evolution of quantum computers created various candidates for secure cryptographic schemes based on the multivariate quadratic polynomial problem [8]. These type of cryptosystems are considered to be an active issue of research due to their quantum resilience.

### 3.4. Lattice-Based Cryptosystems

Cryptographic algorithms that are based on lattice theory have gained the interest of researchers and are perhaps the most famous of all candidates for post-quantum cryptography. Imagine a lattice like a set of points in an *n* dimensional space with periodic structure. The algorithms which are implemented in lattice-based cryptosystems are characterized by simplicity and efficiency and are highly parallelizable [35].

Lattice-based cryptographic protocols are proven to be secure, as their strong security relies on well-known lattice problems such as the Shortest Vector Problem (SVP) and the Learning with Errors problem (LWE) [36]. Additionally, they create powerful and efficient cryptographic primitives, such as functional encryption and fully homomorphic encryption [37]. Moreover, lattice-based cryptosystems create several applications, such as key exchange protocols and digital signature schemes. For all these reasons, lattice-based cryptographic schemes are believed to be the most dynamic field of exploration in post-quantum cryptography and the most prominent and promising one.

## 4. Lattices

Lattices are considered to be a typical subject in both cryptography and cryptanalysis and an essential tool for future cryptography, especially with the transition to the quantum computing era. The study and the analysis of the lattices goes back to the 18th century, when C.F. Gauss and J.L. Lagrange used lattices in number theory and H. Minkowski with his great work "geometry of numbers" sparked the study of lattice theory [38]. In the late 1990s, a lattice was used for the first time in a cryptographic scheme, and in recent years the evolution in this scientific field has been enormous, as there are lattice-based cryptographic schemes for encryption, digital signatures, trapdoor functions, and much more.

A lattice is a discrete subgroup of points in n-dimensional space with periodic structure. Any subgroup of $\mathbb{Z}^n$ is a lattice, which is called integer lattice. It is appropriate to describe a lattice using its basis [35]. The basis of a lattice is a set of independent vectors in $\mathbb{R}^n$ and by combining them, the lattice can be generated.

**Definition 1.** *A set of vectors $\{b_1, b_2, \ldots, b_n\} \subset \mathbb{R}^m$ is linearly independent if the equation*

$$c_1 b_1 + c_2 b_2 + \cdots + c_n b_n = 0, where\ c_i \in \mathbb{R}(i = 1, \ldots, n)$$

*accepts only the trivial solution $c_1 = c_2 = \cdots = c_n = 0$.*

**Definition 2.** *Given n linearly independent vectors $b_1, b_2, \ldots, b_n \in R^m$, the lattice generated by them is defined as*

$$\mathcal{L}(b_1, b_2, \ldots, b_n) = \{\sum x_i b_i / x_i \in \mathbb{Z}\}.$$

*Therefore, a lattice consists of all integral linear combinations of a set of linearly independent vectors and this set of vectors $\{b_1, b_2, \ldots, b_n\}$ is called a lattice basis. So, a lattice can be generated by different bases as can be seen in Figure 1.*

**Definition 3.** *The same number $dim(\mathcal{L})$ of elements of all the bases of a lattice $\mathcal{L}$ it is called the dimension (or rank) of the lattice, since it matches the dimension of the vector subspace $span(\mathcal{L})$ spanned by $\mathcal{L}$.*

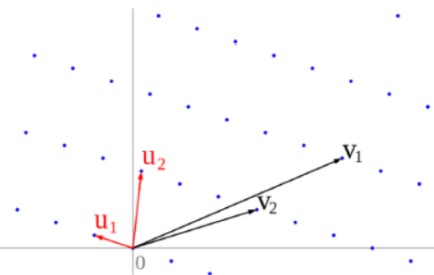

**Figure 1.** Bases of a lattice.

**Definition 4.** *Let $\mathcal{L}$ be a lattice with dimension n and $B = \{b_1, b_2, \ldots, b_n\}$ a basis of the lattice. We define as fundamental parallelepiped the set:*

$$\mathcal{P}(b_1, b_2, \ldots, b_n) = \{t_1 b_1, t_2 b_2, \ldots, t_n b_n : 0 \le t_i < 1\} = \sum_{j=1}^{n} [0, 1) b_j$$

Not every given set of vectors forms a basis of a lattice and the following theorem gives us a criterion.

**Theorem 1.** *Let $\mathcal{L}$ be a lattice with rank n and $\{b_1, b_2, \ldots, b_n\} \in \mathcal{L}$, n linearly independent lattice vectors. The vectors $\{b_1, b_2, \ldots, b_n\}$ form a basis of $\mathcal{L}$ if and only if $\mathcal{P}(b_1, b_2, \ldots, b_n) \cap \mathcal{L} = \{0\}$.*

**Definition 5.** *A matrix $U \in \mathbb{Z}^{n \times n}$ is called unimodular if $det U = \pm 1$.*
*For example, the matrix*

$$\begin{pmatrix} 4 & 5 \\ 13 & 16 \end{pmatrix}$$

*with $det(U) = -1$.*

**Theorem 2.** *Two bases $B_1, B_2 \in \mathbb{R}^{m \times n}$ generate the same lattice if and only if there is an unimodular matrix $U \in \mathbb{R}^{n \times n}$ such that $B_2 = B_1 U$.*

**Definition 6.** *Let $\mathcal{L} = \mathcal{L}(\mathcal{B})$ be a lattice of rank n and let B a basis of $\mathcal{L}$. We define the determinant of $\mathcal{L}$ denoted $det(\mathcal{L})$, as the n-dimensional volume of $\mathcal{P}(\mathcal{B})$.*
*We can write*

$$det(\mathcal{L}(\mathcal{B})) = vol(P) \text{ and also}$$

$$det(\mathcal{L}) = \sqrt{det(B^T B)}.$$

An interesting property of the lattices is that the smaller the determinant of the lattice is, the denser the lattice is.

**Definition 7.** *For any lattice $\mathcal{L} = \mathcal{L}(\mathcal{B})$, the minimum distance of $\mathcal{L}$ is the smallest distance between any two lattice points:*

$$\lambda(\mathcal{L}) = inf\{\|x - y\| : x, y \in \mathcal{L} / x \ne y\}$$

It is obvious that the minimum distance can be equivalently defined as the length of the shortest nonzero lattice vector:

$$\lambda(\mathcal{L}) = inf\{\|v\| : v \in \mathcal{L}, \{0\}\}$$

*4.1. Shortest Vector Problem (SVP)*

The Shortest Vector Problem (SVP) is a very interesting and extensively studied computational problem on lattices. The Shortest Vector Problem states that given a lattice $\mathcal{L}$, the shortest nonzero vector in $\mathcal{L}$ should be found.

That is to say, given a basis $B = \{b_1, b_2, \ldots, b_n\} \in \mathbb{R}^{m \times n}$, the shortest vector problem is to find a vector $\vec{v}$ satisfying

$$\|\vec{v}\| = \min_{\vec{u} \in \mathcal{L}(\mathcal{B}) \ /0} = \lambda(\mathcal{L}(\mathcal{B}))$$

which is a variant of the Shortest Vector Problem is computing the length of the shortest nonzero vector in $\mathcal{L}$ (e.g., $\lambda(\mathcal{L})$) without necessarily finding the vector.

**Theorem 3.** *Minkowski's first theorem. The shortest nonzero vector in any n-dimensional lattice $\mathcal{L}$ has length at most $\gamma_n det(\mathcal{L})^{1/n}$, where $\gamma_n$ is an absolute constant (approximately equals to $\sqrt{n}$) that depend only of the dimension n and $det(\mathcal{L})$ is the determinant of the lattice.*

Two great mathematicians, J. Lagrange and C.F. Gauss, were the first ones to study the lattices and they knew an algorithm to find the shortest nonzero vector in two dimensional lattices. In 1773, Lagrange proposed an efficient algorithm to find a shortest vector of a lattice and Gauss, working independently, made a publication with his proposal for this algorithm in 1801 [38].

A $g$-approximation algorithm for SVP is an algorithm that on input a lattice $\mathcal{L}$, outputs a nonzero lattice vector of length at most $g$ times the length of the shortest vector in the lattice. The LLL lattice reduction algorithm is capable of approximating SVP within a factor $g = O((2/\sqrt{3})n)$ where $n$ is the dimension of the lattice. Micciancio proved that the Shortest Vector Problem is NP-hard even to approximate within any factor less than $\sqrt{2}$ [39]. SVP is considered to be a hard mathematical problem and can be used as a cornerstone for the construction of provably secure cryptographic schemes, such as lattice-based cryptography.

One more form of the CVP is figuring the distance of the objective from the lattice without finding the nearest vector of the lattice, and numerous applications are only interested in finding a vector in the lattice that is somewhat close to the objective, not necessarily the nearest one.

*4.2. Closest Vector Problem (CVP)*

The Closest Vector Problem (CVP) is a computational problem on lattices that relates closely to the Shortest Vector Problem. CVP states that given a target point $\vec{x}$, the lattice point closest to the target should be found.

Let $\mathcal{L}$ be a lattice and a fixed point $t \in \mathbb{R}^n$; we define the distance:

$$d(t, \mathcal{L}) : min_{x \in \mathcal{L}} \|x - t\|.$$

CVP can be formulated as follows: Given a basis matrix $B$ for the lattice $\mathcal{L}$ and a $t \in \mathbb{R}^n$, compute a non-zero vector $v \in \mathcal{L}$ such that $\|t - v\|$ is minimal. So, we search a non-zero vector $v \in \mathcal{L}$, such that $\|v\| = d(t, \mathcal{L})$.

Another version of the CVP is computing the distance of the target from the lattice without finding the closest vector of the lattice, and many applications only demand to find a lattice vector that is not too far from the target, not necessarily the closest one [40].

The most famous polynomial-time algorithms to solve the Shortest Vector Problem are Babai's algorithm and Kannan's algorithm which are based on lattice reduction. Below, in Algorithm 1, we present the first algorithm which was proposed by Lazlo Babai in 1986 [41].

---

**Algorithm 1** Babai's Round-off Algorithm.

---

Input: basis $B = \{b_1, b_2, \ldots, b_n\} \in \mathbb{Z}^n$, target vector $c \in \mathbb{R}$
Output: approximate closest lattice point of $c$ in $L(B)$
1: procedure RoundOff
2: Compute inverse of $B : B^{-1} \in \mathbb{Q}^n$
3: $v := B[B^{-1}c]$
4: return $v$
5: end procedure

---

CVP is the foundation of many cryptographic schemes of lattice cryptography, where the decryption procedure corresponds to a CVP computation. It is regarded as NP-hard to solve approximately within any constant factor [42]. Besides cryptography, the problem of finding a good CVP approximation algorithm with approximation factors that grow as a polynomial in the dimension of a lattice has numerous applications in computer science and is an active open problem in lattice theory.

*4.3. Lattice Reduction*

Lattice reduction, or lattice basis reduction, is about finding an interesting, useful basis of a lattice. Such a requested useful basis, from a mathematical point of view, satisfies a few strong properties. A lattice reduction algorithm is an algorithm that takes as input a basis of the lattice and returns a simpler basis which generates the same lattice. For computing science, we are interested in computing such bases in a reasonable time, given an arbitrary basis. In general, a reduced basis is composed from vectors with good properties, such as being short or being orthogonal.

A polynomial-time basis reduction algorithm developed by Laszlo Lovasz, Arjen Lenstra, and Hendrik Lenstra was published in 1982, the LLL, which took its name from the initials of their surnames [43]. The basis reduction algorithm approaches the solution of the smallest vector problem in small dimensions, especially in two dimensions; the shortest vector is too small to be computed in a polynomial time. On the contrary, in large dimensions there is no algorithm known which solves the SVP in a polynomial time. With the aid of the Gram–Schmidt orthonormalization method, we define the base reduction method LLL.

**5. The NTRU Cryptosystem**

A public key cryptosystem known as NTRU was presented in 1996 by Joseph H. Silverman, Jill Pipher, and Jeffrey Hoffstein. [44]. Until 2013, the NTRU cryptosystem was only commercially available, but after that it was released into the public domain for public use. The NTRU is based on the shortest vector problem in a lattice and is one of the fastest public key cryptographic schemes. It encrypts and decrypts data using polynomial rings. NTRU is more efficient than other current cryptosystems such as RSA, and it is believed to be resistant to quantum computer attacks, and this makes it a prominent post-quantum cryptosystem.

To describe the way the NTRU cryptographic scheme operates, we first have to give some definitions.

**Definition 8.** *Fix a positive integer N. The ring of convolution polynomials (of rank N) is the quotient ring*

$$R = \frac{\mathbb{Z}[X]}{(X^N - 1)}. \tag{1}$$

**Definition 9.** *The ring of convolution polynomials (modulo q) is the quotient ring*

$$R_q = \frac{(\mathbb{Z}/q\mathbb{Z})[x]}{(X^N - 1)}. \tag{2}$$

**Definition 10.** *We consider a polynomial $a(x)$ as an element of $R_q$ by reducing its coefficients mopulo q. For any positive integers $d_1$ and $d_2$, we let*

$$\mathcal{L}(d_1, d_2) = a(x) \in R : \left\{ \begin{array}{l} a(x) \text{ has } d_1 \text{ coefficients equal to } 1 \\ a(x) \text{ has } d_2 \text{ coefficients equal to } -1 \\ a(x) \text{ has all other coefficients equal to } 0 \end{array} \right\} \quad (3)$$

*Polynomials in $\mathcal{L}(d_1, d_2)$ are called ternary (or trinary) polynomials. They are analogous to binary polynomials, which have only 0's and 1's as coefficients.*

We assume we have two polynomials $a(x)$ and $b(x)$. The product of these two polynomials is given by the formula

$$a(x) * b(x) = c(x) \text{ with } c_k = \sum_{i=0}^{k} a_i b_{k-i} + \sum_{i=k+1}^{N-1} a_i b_{N+k-i} = \sum_{i+j \equiv k \bmod N} a_i b_j \quad (4)$$

We will denote the inverses by $F_q$ and $F_p$, such that

$$F_q * f \equiv 1 \pmod{q} \text{ and } F_p * f \equiv 1 \pmod{p} \quad (5)$$

*5.1. Description*

The NTRU cryptographic scheme is based firstly on three well-chosen parameters $(N, p, q)$, such that $N$ is a fixed positive large integer, $p$ and $q$, is not necessary to be prime but are relatively prime, e.g., $gcd(p, q) = 1$ and $q$ will be always larger than $p$ [44]. Secondly, NTRU depends on four sets of polynomials $\mathcal{L}_f$, $\mathcal{L}_g$, $\mathcal{L}_\phi$ and $\mathcal{L}_m$ with integer coefficients of degree $N-1$ and works on the ring $R = \frac{\mathbb{Z}[X]}{X^N - 1}$.

Every element $f \in R$ is written as a polyonomial or as vector $f = \sum_{N-1}^{i=0} f_i x^i = [f_0, f_1, \ldots, f_{N-1}]$. We make the assumption that Alice and Bob are the two parties that they want to transfer data, to communicate with security. A trusted party or the first party selects public parameters $(N, p, q, d)$ such that $N, p$ are prime numbers, $gcd(p, q) = gcd(N, q) = 1$ and $q > (6d + 1)p$.

- Alice chooses randomly two polynomials $f(x) \in \mathcal{L}_{(d+1, d)}$ and $g(x) \in \mathcal{L}_{(d, d)}$. These two polynomials are Alice's private key.
- Alice computes the inverse polynomials

$$F_q(x) = f(x)^{-1} \in R_q \text{ and } F_p(x) = f(x)^{-1} \in R_p \quad (6)$$

- Alice computes $h(x) = F_q(x) * g(x) \in R_q$ and the polynomial $h(x)$ is Alice's public key. Alice's private key is the pair $(f(x), F_p(x))$ and by only using this key, she can decrypt messages. Otherwise, she can store it, which is probably $\bmod q$ and compute $F_p(x)$ when she needs it.
  Alice publishes her key $h$.
- Bob wants to encrypt a message and chooses his plaintext $m(x) \in R_p$. The $m(x)$ is a polynomial with coefficients $m_i$ such that $-\frac{1}{2}p \leq m_i \leq \frac{1}{2}p$.
- Bob chooses a random polynomial $r(x) \in \mathcal{T}(d, d)$, which is called ephemeral key, and computes

$$e(x) \equiv ph(x) * r(x) + m(x) \pmod{q} \quad (7)$$

and this is the encrypted message that Bob sends to Alice.

- Alice computes

$$a(x) \equiv f(x) * e(x) \pmod{q} \quad (8)$$

- Alice chooses the coefficients of $a$ in the interval from $-q/2$ to $q/2$ (center lifts $a(x)$ to an element of $R$).

- Alice computes

$$b(x) \equiv F_p(x) * a(x) \,(\mathrm{mod}\, p) \qquad (9)$$

and she recovers the message $m$ as if the parameters have been chosen correctly; the polynomial $b(x)$ equals the plaintext $m(x)$.

Depending on the choice of the ephemeral key $r(x)$ the plaintext $m(x)$ can be encrypted with many ways, as its possible encryptions are $ph(x) * r(x) + m(x)$. The ephemeral key should be used one time only, e.g., it should not be used to encrypt two different plaintexts. Additionally, Bob should not encrypt the same plaintext by using two different ephemeral keys.

*5.2. Discrete Implementation*

- Assume the trusted party chooses the parameters $(N, p, q, d) = (11, 3, 61, 2)$. As we can see, $N = 11$ and $p = 3$ are prime numbers, $gcd(3, 61) = gcd(11, 2) = 1$ and the condition $q > (6d + 1)p$ is satisfied as it is $61 > (6 \cdot 2 + 1)3 = 39$.
- Alice chooses the polynomials

$$f(x) = x^{10} - x^8 - x^6 + x^4 + x^2 + x + 1 \in \mathcal{L}(3, 2)$$
$$g(x) = x^9 - x^8 - x^6 + x^4 + x^2 + 1 \in \mathcal{L}(2, 2)$$

These polynomials, $f, g$ are the private key of Alice.

- Alice computes the inverses

$$F_{61}(x) = f(x)^{-1} \bmod 61 =$$
$$= 45x^{10} + 49x^9 + 26x^8 + 40x^7 + 53x^6 + 47x^5 + 21x^4 + 24x^3 + 60x^2 + 32x + 31 \in R_{61}$$
$$F_3(x) = f(x)^{-1} = x^9 + x^7 + x^5 + 2x^4 + 2x^3 + 2x^2 + x \in R_3$$

Alice can store $(f(x), F_3(x))$ as her private key.

- Alice computes

$$h(x) = F_{61}(x) * g(x) =$$
$$= 11x^{10} + 49x^9 + 26x^8 + 46x^7 + 28x^6 + 53x^5 + 31x^4 + 36x^3 + 30x^2 + 5x + 50$$

and publishes her public key $h(x)$.

- Bob decides to encrypt the message $m(x) = x^7 - x^4 + x^3 + x + 1$ and uses the ephemeral key $r(x) = x^9 + x^7 + x^4 - x^3 + 1$.
- Bob computes and sends to Alice the encrypted message

$$e(x) \equiv ph(x) * r(x) + m(x) \,(\mathrm{mod}\, q)$$

that is

$$e(x) = 11x^{10} + 49x^9 + 52x^8 + 35x^7 + 30x^6 + 25x^5 + 35x^4 + 32x^3 + 18x^2 + 56x + 28 \,(\mathrm{mod}\, 61).$$

- Alice receives the ciphertext $e(x)$ and computes

$$f(x) * e(x) =$$
$$= 58x^{10} + 60x^9 + 60x^8 + 4x^7 + 56x^5 + 6x^4 + 55x^2 + 3x + 6 \in R_{61}$$

- Therefore, Alice centerlifts modulo 61 to obtain

$$a(x) = -3x^{10} - x^9 - x^8 + 4x^7 + 5x^5 + 6x^4 - 6x^2 + 3x + 6 \in R_{61}$$

- She reduces $a(x)$ modulo 3 and computes

$$F_3(x) * a(x) = x^7 + 2x^4 + x^3 + x + 1 \in R_3$$

and recovers Bob's message $m(x) = x^7 - x^4 + x^3 + x + 1$

*5.3. Security*

Lattice-based NTRU is one of the fastest public key cryptosystems and it is used for encryption (NTRU-Encrypt) and digital signatures (NTRUSign). From the moment that NTRU was presented in 1996, NTRU security has been a main issue of interest and research. NTRU hardness relies on the hard mathematical problems in a lattice, such as the Shortest Vector Problem [35].

The authors of NTRU in their paper [44] argue that the secret key can be recovered by the public key, by finding a sufficiently short vector of the lattice that is generated in the NTRU algorithm. D. Coppersmith and A. Shamir proposed a simple attack against the NTRU cryptosystem. In their work, they argued that the target vector $f||g \in \mathbb{Z}^{2N}$ (the symbol $||$ denotes vector concatenation) belongs to the natural lattice:

$$L_{CS} = \{F||G \in \mathbb{Z}^{2N}|F \equiv h * G \bmod q \text{ where } F, G \in \mathbb{R}\}.$$

It is obvious that $L_{CS}$ is a full dimension lattice in $\mathbb{Z}^{2N}$, with volume $q^N$. The target vector is the shortest vector of $L_{CS}$, so the private keys should be outputted heuristically by SVP-oracle $f$ and $g$. Hoffstein et al. claimed that if one chooses the number $N$ reasonably, the NTRU is sufficiently secure, as all these types of attacks are exponential in $N$. These types of attacks are based on the difficulty of solving certain lattice problems, such as SVP and CVP [45]. Lattice attacks can be used to recover the private key of an NTRU system, but they are generally considered to be infeasible for the current parameters of NTRU. It is important that the key size of the NTRU protocol is $O(N \log q)$ and this fact makes NTRU a promising cryptographic scheme for post-quantum cryptography [46].

Furthermore, the cryptanalysis of NTRU is an active area of research and other types of attacks against the NTRU cryptosystem have been developed [47–49]. We refer to some of them as detailed below.

- Brute-Force Attack. In this type of attack, all possible values of the private key are tested until the correct one is found. Brute-force attacks are generally not practical for NTRU, as the size of the key space is very large [50].
- Key Recovery Attack. This type of attack relies on exploiting vulnerabilities in the key-generation process of NTRU. For example, if assuming the arbitrary number generator used to create the confidential key is frail, a fraudulent user may be able to recover the private key [51].
- Side-channel Attack. This type of attack take advantage of the weaknesses in the implementation of NTRU, such as timing attack, power analysis attack, and fault attack. Side-channel attacks require the device to be physically accessible running the implementation [52,53].

To protect NTRU against these types of attacks and avoid the leak of secret data and information, researchers use various techniques to ensure its security, such as parameter selection, randomization, and error-correcting codes.

## 6. The LWE Cryptosystem

In 2005, O. Regev presented a new public key cryptographic scheme, the Learning with Errors cryptosystem, and for this work, Regev won the Godel Prize in 2018 [54]. LWE is one of the most famous lattice-based cryptosystems and one of the most widely studied in recent years. It is based on the Learning with Errors problem and the hardness of finding a random linear function of a secret vector modulo a prime number. A probabilistic cryptosystem with a high probability algorithm is the LWE public key cryptosystem. Since LWE proved to be secure and efficient, it has become one of the most contemporary and innovative research topics in both lattice-based cryptography and computer science.

*6.1. The Learning with Errors Problem*

Firstly, we have to introduce the Learning with Errors problem (LWE). Assuming that we have a secret vector $s = (s_1, s_2, \ldots, s_n) \in \mathbb{Z}^n$ with coefficient integer numbers and $n$ linear equations, such that

$$a_{11}s_1 + a_{12}s_2 + \cdots + a_{1n}s_n \approx a$$
$$a_{21}s_1 + a_{22}s_2 + \cdots + a_{2n}s_n \approx b$$
$$\vdots$$
$$a_{m1}s_1 + a_{m2}s_2 + \cdots + a_{mn}s_n \approx m$$

We use the symbol "$\approx$" to claim that within a certain error, the value approaches the actual response. This is a difficult problem because adding and multiplying rows together will increase the number of errors in each equation, resulting in the final row reduced state being worthless and the answer being far away from the real value.

**Definition 11.** *Let $s \in \mathbb{Z}_q^n$ be a secret vector and $\chi$ be a given distribution on $\mathbb{Z}_q$. An LWE distribution $A_{s,n,q,\chi}$ generates a sample $(a, b) \in \mathbb{Z}_q^n \times \mathbb{Z}_q$ or $(A, b) \in \mathbb{Z}_q^{m \times n} \times \mathbb{Z}_q^m$ where $a \in \mathbb{Z}_q^n$ is uniformly distributed and $b = \langle a, s \rangle + e$, where $e \leftarrow \chi$ and $\langle a, s \rangle$ is the inner product of $a$ and $s$ in $\mathbb{Z}_q$.*

We call $A_{s,n,q,\chi} = (a, b) \in \mathbb{Z}_q^n \times \mathbb{Z}_q$ the LWE distribution, $s$ is called the private key, and $e$ is called the error distribution. If $b \in \mathbb{Z}_q$ is uniformly distributed, then it is called the uniform LWE distribution.

**Definition 12.** *Fix $n \geq 1$, $q \geq 2$ and an error probability distribution $\chi$ on $\mathbb{Z}_q$. Let $s$ be a vector with n coefficients in $\mathbb{Z}_q$. Let $A_{s,\chi}$ on $\mathbb{Z}_q^n \times \mathbb{Z}_q$ be the probability distribution choosing a vector $a \in \mathbb{Z}_q$ uniformly at random, choosing $e \in \mathbb{Z}_q$ according to $\chi$ and outputting $(a, \langle a, s \rangle + e)$ where additions are performed in $\mathbb{Z}_q$. We say an algorithm solves LWE with modulus q and error distribution $\chi$ if for any $s \in \mathbb{Z}_q^n$ given enough samples from $A_{s,\chi}$ it outputs $s$ with high probability.*

**Definition 13.** *Suppose we have a way of generating samples from $A_{s,\chi}$ as above, and also generating random uniformly distributed samples of $(a, b)$ from $\mathbb{Z}_q^n \times \mathbb{Z}_q$. We call this uniform distribution $U$. The decision-LWE problem is to determine after a polynomial number of samples whether the samples are coming from $A_{s,\chi}$ or $U$.*

Simplifying the definition and formulated in more compact matrix notation, if we want to generate a uniformly random matrix $A$ with coefficients between 0 and $q$ and two secret vectors $s, e$ with coefficients drawn from a distribution with small variance, the LWE sample can be calculated as: $(A, b = As + e \bmod q)$. According to the LWE problem, it is challenging to locate the secret $s$ from such a sample.

**Definition 14.** *For $a > 0$, the family $\Psi_a$ is the (uncountable) set of all elliptical Gaussian distributions $D_r$ over a number field $K_\mathbb{R}$ in which $r \geq a$.*

The choice of the parameters is crucial for the hardness of this problem. The distribution is a Gaussian distribution or a binomial distribution with variance 1 to 3; the length of the secret vector $n$ is such that $2^9 < n < 2^{10}$ and the modulus $q$ is in the range $2^8$ to $2^{16}$.

*6.2. Description*

Assume $n \geq 1$, $q \geq 2$ are positive integers and $\chi$ is a given probability distribution in $\mathbb{Z}_q$. The LWE cryptographic scheme is based on LWE distribution $A_{s,\chi}$ and is described below.

The parameters of the LWE cryptosystem are crucial to the protocol's security. So, let $n$ be the security parameter of the system; $m, q$ are two integer numbers and $\chi$ is a probability distribution on $\mathbb{Z}_q$.

The security and the correctness of the cryptosystem are based on the following parameters, which are be chosen appropriately.

- Choose $q$, a prime number between $n^2$ and $2n^2$.
- Let $m = (1 + \epsilon)(n + 1) \log q$ for some arbitrary constant $\epsilon > 0$.
- The probability distribution is chosen to be $\chi = \Psi_{a(n)}$ for $a(n) \in O(1/\sqrt{n} \log n)$

We suppose that there are two parties, Alice and Bob, who want to transfer information securely. The LWE cryptosystem has the typical structure of a cryptographic scheme and its steps are the following.

- Alice chooses uniformly at random $s \in \mathbb{Z}_q^n$. $s$ is the private key.
- Alice generates a public key by choosing $m$ vectors $a_1, a_2, \ldots, a_m \in \mathbb{Z}_q^n$ independently from the uniform distribution. She also chooses elements (error offsets) $e_1, e_2, \ldots, e_m \in \mathbb{Z}_q^n$ independently according to $\chi$. The public key is $(a_i, b_i)_{i=1}^m$, where $b_i = \langle a_i, s \rangle + e_i$. In matrix form, the public key is the LWE sample $(A, b = As + e \mod q)$, where $s$ is the secret vector.
- Bob, in order to encrypt a bit, chooses a random set $S$ uniformly among all $2^m$ subsets of $[m]$. The encryption is $(\sum_{i \in S} a_i, \sum_{i \in S} b_i)$ if the bit is 0 and $(\sum_{i \in S} a_i, \lfloor \frac{q}{2} \rfloor + \sum_{i \in S} b_i)$ if the bit is 1.

  In matrix form, Bob can encrypt a bit $m$ by calculating two LWE problems : one using $A$ as random public element, and one using $b$. Bob generates his own secret vectors $s', e'$ and $e$ and make the LWE samples $(A, b' = A^T s' + e' \mod q)$, $(b, v' = b^T s' + e'' \mod q)$. Bob has to add the message that wants to encrypt to one of these samples, where $v'$ is a random integer between 0 and $q$. The encrypted message of Bob consists of the two samples $(A, b' = A^T s' + e' \mod q)$, $(b, v' = b^T s' + e'' + \frac{q}{2} m \mod q)$ .
- Alice wants to decrypt Bob's ciphertext. The decryption of a pair $(a, b)$ is 0 if $b - \langle a, s \rangle$ is closer to 0 than to $\lfloor \frac{q}{2} \rfloor$ modulo $q$. In another case, the decryption is 1.

  In matrix form, Alice firstly calculates $\Delta v = v' - b'^T s$. As long as $e^T s' + e'' - s^T e'$ is small enough, Alice recovers the message as $mes = \lfloor \frac{2}{q} \Delta v \rceil$.

### 6.3. Discrete Implementation

We choose $n = 4$ and $q = 13$.

- Alice chooses the private key $s = [2, 5, 0, 6]$.
- Let $m = 3$ so Alice generates the public key with the aid of three vectors $a_i, i = 1, 2, 3$ and three elements $e_i, i = 1, 2, 3$ (error terms). She chooses : $a_1 = [1, 6, 2, 4]$ and $e_1 = 1$, $a_2 = [0, 3, 5, 1]$ and $e_2 = 0$ and $a_3 = [2, 1, 6, 3]$ and $e_3 = -1$. Therefore, Alice's public key is:

$$\{([1, 6, 2, 4], 4), ([0, 3, 5, 1], 8), ([2, 1, 6, 0], 0)\}$$

- Bob wants to encrypt 0 so he takes the subset $S = \{1, 2\}$. So, he computes

$$\left(\sum_{i \in S} a_i, \sum_{i \in S} b_i\right) = ([1, 6, 2, 4] + [0, 3, 5, 1], 4 + 8) = ([1, 9, 7, 5], 12)$$

- Alice performs the decryption algorithm by computing

$$b - \langle a, s \rangle = 12 - \langle [1, 9, 7, 5], [2, 5, 0, 6] \rangle = 12 - 12 = 0$$

and obviously the decryption is 0 since the output value is closer to 0 (in this case equal to 0) than to $\lfloor \frac{13}{2} \rfloor$ modulo 13.

Therefore, the encryption scheme worked correctly.

### 6.4. Implementations and Variants

The Learning with Errors (LWE) cryptosystem is a popular post-quantum cryptographic scheme that relies on the hardness of using lattices to solve particular computational problems. There are several variants of the LWE cryptosystem, including the

Ring-LWE, the Dual LWE, the Module-LWE, the Binary-LWE, the multilinear LWE, and others [55–57].

The RING-LWE Cryptosystem

This variant of LWE uses polynomial rings instead of the more general lattices used in standard LWE. Ring-LWE has a simpler structure, which improves execution speed and memory utilization efficiency. In 2013, Lyubashevsky et al. [46] presented a new public key cryptographic scheme that is based in the LWE problem.

The Ring-LWE cryptosystem structure.

Lyubachevsky et al. proposed a well-analyzed cryptosystem that uses two ring elements for both public key and ciphertext and it is a plain lattice-based version of the public key cryptographic system.

The two parties they want to communicate agree on the complexity value of $n$, the highest co-efficient power to be used. Let $R = \frac{\mathbb{Z}[X]}{(X^n + 1)}$ be the fixed ring and an integer $q$ is chosen, such as $q = 2n - 1$. The steps of the Ring-LWE protocol are described below.

- A secret vector $s$ with $n$ length is chosen with modulo $q$ integer entries in ring $R_q$, where $q \in \mathbb{Z}^+$. This is the private key of the system.
- An element $a \in R_q$ is chosen and a random small element $e \in R$ from the error distribution and we compute $b = a\dot{s} + e$.
  The public key of the system is the pair $(a, b)$.
- Let $m$ be the $n$ bit message that is meant for encryption.
  1. The message $m$ is considered an element of $R$ and the bits are used as coefficients of a polynomial of a degree less than $n$.
  2. The elements $e_1, e_2, r \in R$ are generated from error distribution.
  3. The $u = a \cdot r + e_1 \bmod q$ is computed.
  4. The $v = b \cdot r + e_2 + \cdot \lfloor \frac{q}{2} \rfloor \cdot m \bmod q$ is computed and it is sent $(u, v) \in R_q^2$ to receiver.
- The second party receives the payload $(u, v) \in R_q^2$ and computes $r = v - u \cdot s = (r \cdot e - s \cdot e_1 + e_2) + \lfloor \frac{q}{2} \rfloor \cdot m \bmod q$. Each $r_i$ is evaluated and if $r_1 \approx \frac{q}{2}$, then the bits are recovered back to 1, or else 0.
  The Ring-LWE cryptographic scheme is similar to the LWE cryptosystem that was proposed by Regev. Their difference is that the inner products are replaced with ring products, so the result is a new ring structure, increasing the efficiency of the operations.

*6.5. Security*

Learning with Errors (LWE) is a computational problem that is the basis for cryptosystems and especially for cryptographic schemes of post-quantum cryptography. It is considered to be a hard mathematical problem and as a consequence, cryptosystems that are based on the LWE problem are of high security as well. LWE cryptographic protocols are a contemporary and active field of research and therefore their security is studied and analyzed continually and steadily.

There are various attacks that can be performed against the cryptosystems which are based on the LWE problem. We can say that these types of attacks are, in general, attacks that exploit weaknesses in the LWE problem itself, and attacks that exploit weaknesses in the specific implementation of the cryptosystem. Below, we present some of these types of attacks that can be launched against LWE-based cryptographic schemes.

- Dual Attack. This type of attack is based on the dual lattice and is most effective against LWE instances with small size of plaintext messages.
  Thus, hybrid dual attacks are appropriate for spare and small secrets, and in a hybrid attack, one estimates part of the secret without knowledge and performs some attacks on the leftover part [58] The cost of attacking the remaining portion of the secret is decreased because guessing reduces the problem's size. Additionally, the component

of the lattice attack can be utilized for multiple guesses. When the lattice attack component is a primal attack, we call it a hybrid primal attack and a hybrid dual attack, respectively, and the optimal attack is achieved when the cost of guessing is equal to the lattice attack cost.

- Sieving Attack. This type of attack relies on the idea of sieving, which claims to find linear combinations of the LWE samples that reveal information about the secret. Sieving attacks can be used to solve the LWE problem with fewer samples than its original complexity.
- Algebraic attack. This type of attack is based on the idea of finding algebraic relations between the LWE samples that let out secret data information. Algebraic attacks can be suitable for solving the LWE problem with fewer samples than the original complexity as well.
- Side-channel attack. This type of attack exploits weaknesses in the implementation of the LWE-based scheme, such as timing attacks and others. Side-channel attacks are generally easier to mount than attacks against the LWE problem itself, but they require physical access to the device running the implementation.
- Attacks that use the BKW algorithm. This is a classic attack; it is considered to be sub-exponential and is most effective against small or small-structured LWE instances.

To mitigate these attacks, LWE-based schemes typically use various techniques such as parameter selection, randomization, and error-correcting codes. These techniques are designed to make the LWE problem harder to solve and to prevent attackers from taking advantage of vulnerabilities in the implementation [59,60].

## 7. The GGH Cryptosystem

In 1997, Oded Goldreich, Shafi Goldwasser, and Shai Halevi proposed a cryptosystem (GGH) [61] based on algrebraic coding theory and it can be seen as a lattice analogue of the McEliece cryptosystem [29]. In both the GGH and McEliece schemes, the addition of a random noise vector to the plaintext is called the ciphertext [35]. In the GGH cryptosystem, the public and the private key are a representation of a lattice and in the McEliece, the public and the private key are a representation of a linear code. The basic distinction between these two cryptographic schemes is that the domains in which the operations take place are different. The main idea and structure of the GGH cryptographic scheme is characterized by simplicity and it is based on the difficulty of reducing lattices.

### 7.1. Description

The GGH public key encryption scheme is formed by the key generation algorithm $K$, the encryption algorithm $E$, and the decryption algorithm $D$. It is based on lattices in $\mathbb{Z}^n$ , a key derivation function $h : \mathbb{Z}^n \times \mathbb{Z}^n \to K_s$ and a symmetric cryptosystem $(K_s, P, C, E_s, D_s)$, where $K$ is the key generation algorithm, $P$ the set of plain texts, $C$ the set of ciphertexts, $E_s$ the encryption algorithm, and $D_s$ the decryption algorithm.

- The key generation algorithm $K$ generates a lattice $L$ by choosing a basis matrix $V$ that is nearly orthogonal. An integer matrix $U$ it is chosen which has determinant $det(U) = \pm 1$ and the algorithm computes $W = UV$. Then, the algorithm outputs $ek = W$ and $dk = V$.
- The encryption algorithm $E$ receives as input an encryption key $ek = W$ and a plain message $m \in P$. It chooses a random vector $u \in \mathbb{Z}^n$ and a random noise vector $u$. Then it computes $x = uW$, $z = x + r$ and encrypts the message $w = E_s(h(x,r), m)$. It outputs the ciphertext $c = (z, w)$.
- The decryption algorithm $D$ takes as input a decryption key $dk = V$ and a ciphertext $c = (z, w)$. It computes $x = \lfloor zV^{-1} \rceil V$ and $r = z - x$ and decrypts as $m = D_s(h(x,r), w)$. If $D_s$ algorithm outputs the symbol $\perp$ the decryption fails and then $D$ outputs $\perp$, otherwise the algorithm outputs $m$.

We assume that there exist two users, Alice and Bob, who want to communicate secretly. The main (classical) process of the GGH cryptosystem is described below.

1.  Alice chooses a set of linearly independent vectors $v_1, v_2, \ldots, v_n \in \mathbb{Z}^n$ which form the matrix $V = [v_1, v_2, \ldots, v_n], v_i \in \mathbb{Z}^n, 1 \leq i \leq n$. Alice, by calculating the Hadamard Ratio of matrix $V$ and verifying that is not too small, checks her vector's choice. This is Alice's private key and we let $L$ be the lattice generated by these vectors.
2.  Alice chooses an $n \times n$ unimodular matrix $U$ with integer coefficients that satisfies $det(U) = \pm 1$.
3.  Alice computes a bad basis $w_1, w_2, \ldots, w_n$ for the lattice $L$, as the rows of $W = UV$, and this is Alice's public key. Then, she publishes the key $w_1, w_2, \ldots, w_n$.
4.  Bob chooses a plaintext that he wants to encrypt and he chooses a small vector $m$ (e.g., a binary vector) as his plaintext. Then, he chooses a small random "noise" vector $r$ which acts as a random element and $r$ has been chosen randomly between $-\delta$ and $\delta$, where $\delta$ is a fixed public parameter.
5.  Bob computes the vector $e = mW + r = \sum_{i=1}^{n} m_i w_i + r = x_1 w_1 + x_2 w_2 + \cdots + x_n w_n + r$ using Alice's public key and sends the ciphertext $e$ to Alice.
6.  Alice, with the aid of Babai's algorithm, uses the basis $v_1, v_2, \ldots, v_n$ to find a vector in $L$ that is close to $e$. This vector is the $a = mW$, since the "noise" vector $r$ is small and since she uses a good basis. Then, she computes $aW^{-1} = mWW^{-1}$ ans she recovers $m$.

Supposing there is an eavesdropper, Eve, who wants to obtain information of the communication between Alice and Bob. Eve has in her possession the message $e$ that Bob sends to Alice and therefore tries to find the closest vector to $e$, solving the CVP, using the public basis $W$. As she uses vectors that are not reasonably orthogonal, Eve will recover a message $\hat{e}$ which probably will not be near to $m$.

*7.2. Discrete Implementation*

*   Alice chooses a private basis $\vec{v_1} = (48, 1)$ and $\vec{v_2} = (-1, 48)$ which is a good basis since $\vec{v_1}$ and $\vec{v_2}$ are orthogonal vectors, e.g., it is $\langle \vec{v_1}, \vec{v_2} \rangle = 0$. The rows of the matrix $V = \begin{pmatrix} 48 & 1 \\ -1 & 48 \end{pmatrix}$ are Alice's private key. The lattice $L$ spanned by $\vec{v_1}$ and $\vec{v_2}$ has determinant $det(L) = 2305$ and the Hadamard ratio of the basis is $\mathcal{H} = (det(L)/|\vec{v_1}||\vec{v_2}|)^{1/3} \simeq 1$
*   Alice chooses the unimodular matrix $U$ that its determinant is equal to 1, such that $U = \begin{pmatrix} 5 & 8 \\ 3 & 5 \end{pmatrix}$ with $det(U) = +1$.
*   Alice computes the matrix $W$, such that $W = UV = \begin{pmatrix} 232 & 389 \\ 139 & 243 \end{pmatrix}$. Its rows are Alice's bad basis $\vec{w_1} = (232, 389)$ and $\vec{w_2} = (139, 243)$, since it is $cos(\vec{w_1}, \vec{w_2}) \simeq 0.99948$ and these vectors are nearly parallel, so they are suitable for a public key.
*   It is very important for the noise vector to be selected carefully and that it is not shifted where the nearest point is located. For Alice's basis that generates the lattice $L$, $\vec{r}$ is chosen that $|\vec{r}| < 20$. So, the vector $\vec{r}$ is chosen to be $(r_x, r_y)$ with $-10 \leq r_x$ and $r_y \leq 10$.
*   Bob wants to encrypt the message $m = (35, 27)$. The message can be seen as a linear combination of the basis $\vec{w_1}, \vec{w_2}$, such as $35\vec{w_1} + 25\vec{w_2}$ and the noise vector $\vec{r}$ can be added.
*   The corresponding ciphertext is $e = mW + r = (35, 27) \begin{pmatrix} 232 & 389 \\ 139 & 243 \end{pmatrix} + (-9, 1) = (19{,}285, 17{,}064) + (-9, 1) = (19{,}276, 17{,}065)$ and Bob sends it to Alice.
*   Alice, using the private basis, applies Babai's algorithm and finds the closest lattice point. So, she solves the equation $a_1(48, 1) + a_2(-1, 48) = (19{,}276, 17{,}065)$ and finds $a_1 \simeq 463.02$ and $a_2 \simeq 345.8$. So, the closest lattice point is $a_1(48, 1) + a_2(-1, 48) = 463(48, 1) + 346(-1, 48) = (21{,}878, 17{,}071)$ and this lattice vector is close to $e$.
*   Alice realizes that Bob must have computed $(21{,}878, 17{,}071)$ as a linear combination of the public basis vectors and then solving the linear combination again $m_1(232, 389) +$

$m_2(139, 243) = (21,878, 17,071)$, she finds $m_1 = 35$ and $m_2 = 27$ and recovers the message $m = (m_1, m_2) = (35, 27)$.

Eve has in her possession the encrypted message $(19,276, 17,065)$ that Bob had sent to Alice and she tries to solve the CVP using the public basis. So, she is solving the equation $m_1(232, 389) + m_2(139, 243) = (19,276, 17,065)$; she finds the incorrect values $m_1 \simeq 1003.1$, $m_2 \simeq -1535.5$ and recovers the incorrect encryption $m' = (m_1, m_2) = (1003, -1535)$.

In 1999 and in 2001, D. Micciancio proposed a simple technique to reduce both the size of the key and size of the ciphertext of GGH cryptosystem without decreasing the level of its security [62,63].

*7.3. Security*

In the GGH cryptographic scheme, if a security parameter $n$ is chosen, the time required for encryption and the size of the key is $O(n^2 \log n)$ and it is more efficient than other cryptosystems such as AD.

There are some natural ways to perform an attack on the GGH cryptographic scheme.

1.  Leak information and obtain the private key $V$ from the public key $W$.
    For this type of attack, a lattice basis reduction (LLL) algorithm is performed on the public key, the matrix $W$. It is possible that the output is a basis $W'$ that is good enough to enable the effective solution of the necessary instances of the closest vector. It will be extremely difficult for this attack to succeed if the dimension of the lattice is sufficiently large.
2.  Assuming we have a small error vector $r$, try to extract information about the message from the ciphertext $e$.
    For this type of attack, it is useful that in the ciphertext $e = mW + r$, the error vector $r$ is a vector with small entries. An idea is to compute $eW^{-1} = mWW^{-1} + rW^{-1}$ and try to deduce possible values for some entries of $rW^{-1}$. For example, if the $j$-th column of $W^{-1}$ has a particularly small norm, then one can deduce that the $j$-th entry of $rW^{-1}$ is always small and hence get an accurate estimate for the $j$-th entry of $m$. To defeat this attack, one should only use some low-order bits of some entries of $m$ to carry information, or use an appropriate randomized padding scheme
3.  Try to solve the Closest Vector Problem of $e$ with respect to the lattice that is being generated by $W$, for example, by performing the Babai's nearest plane algorithm or the embedding technique.

Moreover, certain types of attacks can be performed against GGH which are discussed below, such as Nguyen's attack and Lee and Hahn attack.

Goldreich, Goldwasser, and Halevi claimed that increasing the key size compensates for the decrease in computation time [35]. When presenting their paper, the three authors published five numerical challenges that corresponded to increase the value of the parameters $n$ in higher dimensions with the aim of supporting their algorithm. In each challenge, a public key and a ciphertext were given and it was requested to recover the plaintext.

In 1999, P. Nguyen exploited the weakness specific to the way the parameters are chosen and developed an attack against the GGH cryptographic scheme [64]. The first four challenges, for $n = 200, 250, 300, 350$ were broken; since then, GGH is considered to be broken partially in its original form. Nguyen argued that the choice of the error vector is its weakness and that it makes it vulnerable to a possible attack. The error vectors used in the encryption of the GGH algorithm must be shorter than the vectors that generate the lattice. This weakness makes Closest Vector Problem instances arising from GGH easier than general CVP instances [35].

The other weakness of the GGH cryptosystem is the choice of the error vector $e$ in the encryption algorithm procedure. The $e$ vector is in $\{\pm\sigma\}^n$ and it is chosen to maximize the Euclidean norm under requirements on the infinity norm. Nguyen takes the ciphertext $c = mB + e$ modulo *sigma*, where $m$ is the plaintext and $B$ the public key, and the $e$ disappears from the equation. This is because $e \in \{\pm\sigma\}^n$ and every choice is 0 mod $\sigma$. So, this leaks information about the message $m \pmod{\sigma}$ and increasing the modulus to $2\sigma$

and adding an all $-\sigma$ vector $s$ to the equation. If this equation is solved for $m$, it leaks information for $m(\text{mod}\,2\sigma)$. Nguyen also demonstrated that in most cases, this equation could be easily solved for $m$.

In 2006, Nguyen and Regev performed an attack on the GGH signatures scheme, transforming a geometrical problem to a multivariate optimization problem [65]. The final numerical challenge for $n = 400$ was solved by M.S. Lee and S.G. Hahn in 2010 [66]. Therefore, GGH has weaknesses and trapdoors, such that it is vulnerable to certain type of attacks, such as one attack that allows a fraudulent user to recover the secret key using a small amount of information about the ciphertext. Specifically, if an attacker can obtain the two smallest vectors in the lattice, they can give information and recover the secret key using Coppersmith's algorithm [67]. As a result, GGH has limited practical use and has been largely superseded by newer and more secure lattice-based cryptosystems. So, while GGH made an important early contribution to the field of lattice-based cryptography, it is not currently considered a practical choice for secure communication due to its limitations in security.

## 8. Evaluation, Comparison and Discussion

We have presented a few of the main cryptographic schemes that are based on the hardness of lattice problems and especially based on the Closest Vector Problem. GGH is a public key cryptosystem which is based in algebraic coding theory. A plaintext is been added with a vector noise and the result of this addition is a ciphertext. Both the private and the public keys are a depiction of a lattice and the private key has a specific structure. Nguyen's attack [64] revealed the weakness and vulnerability of the GGH cryptosystem and many researchers after that considered GGH to be unusable [64,68]

Therefore, in 2010, M.S. Lee and S.G. Hahn presented a method that solved the numerical challenge of the highest dimension 400 [66]. Applying this specific method, Lee and Hann came to the conclusion that the decryption of the ciphertext could be accomplished using partial information of the plaintext. Thus, this method requires some knowledge of the plaintext and cannot be performed in actually real cryptanalysis circumstances. On the other side, in 2012 M. Yoshino and N. Kunihiro and C. Gu et al. in 2015 presented a few modifications and improvements in the GGH cryptosystem, claiming that they made it more resistant to these attacks [67,69].

The same year, C.F. de Barros and L.M. Schechter, in their paper "GGH may not be dead after all", proposed certain improvements for GGH and finally a variation of the GGH cryptographic scheme [70]. De Barros and Schecher, by reducing the public key in order to find a basis with the aid of Babai's algorithm, perform a direct way to attack to GGH. They increase the length of the noise vector $\vec{r}$ setting a new parameter $k$ that modified the GGH cryptographic algorithm. Their modifications resulted in a variation of GGH more resistant to cryptanalysis, but with slower decryption process of the algorithm. In 2015, Brakerski et al. described certain types of attacks against some variations of the GGH cryptosystem and relied on the linearity of the zero-testing procedure [71].

GGH was a milestone in the evolution of post-quantum cryptography; it was one of the earliest lattice-based cryptographic schemes and it is based on the Shortest Vector Problem's difficulty. Even though is is viewed as one of the most significant lattice-based cryptosystems and still has a theoretical interest, it is not recommended for practical use due to its security weaknesses. GGH is less efficient than other lattice-based cryptosystems [72]. The process to encrypt and decrypt a message requires a large amount of computations and this fact makes the GGH cryptosystem obviously slower and less practical than other lattice-based cryptosystems.

Thus, the GGH protocol is vulnerable to certain attacks, such as Coppersmith's attack and Babai's nearest plane algorithm, and it is considered not to be strong enough. These attacks disputed the security of the GGH and made it less preferable than newer, stronger, and more secure lattice-based cryptosystems. Evaluating the efficiency of GGH cryptographic protocol, GGH is relatively inefficient compared to other lattice-based cryp-

tosystems such as NTRU, LWE, and others, and especially in the key generation and for large key length. As the GGH cryptosystem is based in multiplications of matrices, when we choose large keys, it requires a computationally expensive basis reduction algorithm for the encryption and decryption procedure.

Moreover, GGH is considered to be a complex cryptographic scheme which requires concepts and knowledge of lattices and linear algebra to study, analyze, and implement. GGH also has one more drawback, which is the lack of standardization, and this makes hard the comparison of its functionality, security, and connectivity with other cryptographic schemes. GGH was one of the first cryptographic schemes that were developed based on lattice theory and cryptography. In spite of the fact that GGH certainly has interesting theoretical basis and properties, GGH is not used in practice due to its limitations in security, efficiency, and complexity.

NTRU is a public key cryptographic scheme that is based on the Shortest Vector Problem in a lattice and was first presented in the 1990s. It is one of the most well studied and analyzed lattice-based cryptosystems and there have been many cryptanalysis studies of NTRU algorithms, including NTRU signatures. NTRU has a high level of security and efficiency and it is a promising protocol for post-quantum cryptography. Moreover, the NTRU cryptographic algorithm uses polynomial multiplication as its basic operation and it is notable for its simplicity.

A main advantage of the NTRU cryptosystem is its speed and it has been used in certain commercial applications where speed is a priority. NTRU has a fast implementation compared with other lattice-based cryptosystems, such as GGH, LWE, and Ajtai-Dwork. For this reason, NTRU is preferable for applications that require fast encryptions and decryption, such as in IoT devices or in embedded systems. In addition to its speed, NTRU uses smaller key sizes than other public key cryptosystems, but the same level of security is maintained. This makes it ideal for applications or environments with limited memory and processing power.

NTRU is considered to be a secure cryptographic scheme against various types of attacks. It is designed to be resistant against attacks such as lattice basis reduction, meet-in-the-middle attacks, and chosen ciphertext attacks. NTRU is believed to be a strong cryptographic scheme for the quantum era, meaning that it is considered to be resistant against attacks by quantum computers.

NTRU has become famous and widely usable after 2017, because before then, it was under a patent and it was difficult for researchers to use it and modify it. Thus, NTRU is not widely used or standardized in the industry, making it difficult to assess its interoperability with other cryptosystems. Furthermore, NTRU is considered to be a public key cryptographic protocol with relative complexity, and its analysis and implementation require a good understanding of lattice-based cryptography and ring theory. NTRU is a promising lattice-based cryptosystem for post-quantum cryptography that offers fast implementation and strong security guarantees [73].

Learning with Errors (LWE) is a widely used and well-studied public key cryptographic scheme that is based in lattice theory [74]. LWE is considered to be secure against both quantum and classical attacks and indeed, it is considered to be among the most secure and efficient of these schemes, while NTRU has limitations in terms of its security [75]. LWE depends its hardness on the difficulty of finding a random error vector in a matrix product and this makes it a resistant cryptosystem against various types of attacks, the same types of attacks as with NTRU. It is considered to be a strongly secure cryptosystem and post-quantum secure, which means that it is resistant to attacks by a quantum computer [76].

LWE uses keys with small length size compared with other cryptographic schemes that are designed for the quantum era, such as code-based and hash-based cryptosystems [77]. Just like NTRU, LWE is appropriate for implementation in resource-constrained environments, such as in IoT devices or in embedded systems. A basic advantage of the LWE cryptosystem is its flexibility, as it is a versatile cryptographic scheme that can be

suitable in a variety of cryptographic methods such as digital signatures, key exchange, and encryption. LWE also serves as a foundation for more advanced cryptographic protocols, which developed other variations of it.

LWE can be vulnerable to certain type of attacks, such as side-channel attacks, i.e., timing attacks or power analysis attacks, if we do not take the right countermeasures [78]. Just like NTRU, LWE is not considered to be standardized and widely adopted by the computing industry and this makes it difficult to assess its interoperability with other cryptosystems and make a comparison with them. Moreover, LWE cryptographic protocol is characterized by complexity and understanding and modifying it becomes challenging.

Undoubtedly, both NTRU and LWE are fast, efficient, and secure cryptographic schemes. NTRU uses smaller key sizes and that makes it suitable for applications where memory and computational power are limited. Both LWE and NTRU are considered to be strong and resistant to various types of attacks and are considered to be prominent for post-quantum cryptography. Thus, LWE is an adaptable cryptographic protocol and can be used in a wide range of cryptographic tasks and methods, while NTRU is primarily used for encryption and decryption.

In summary, LWE and NTRU are both promising lattice-based cryptosystems that offer strong security guarantees and are resistant to quantum attacks. NTRU is known for its fast implementation and smaller key sizes, while LWE offers more flexibility in cryptographic primitives and is currently undergoing standardization. Ultimately, the choice between LWE and NTRU will depend on specific use cases and implementation requirements.

Overall, each lattice-based cryptosystem has its own strengths and weaknesses depending on the specific use case. Choosing the right one requires careful consideration of factors such as security, efficiency, and ease of implementation.

## 9. Lattice-Based Cryptographic Implementations and Future Research

Quantum research over the past few years has been particularly transformative, with scientific breakthroughs that will allow exponential increases in computing speed and precision. In 2016, the National Institute of Standards and Technology (NIST) announced an invitation to researchers to submit their proposals for developed public—key post-quantum cryptographic algorithms. At the end of 2017, when was the initial submission deadline, 23 signature schemes and 59 encryption—key encapsulation mechanism (KEM) schemes were submitted, in total, 82 candidates' proposals.

In July 2022, the NIST finished the third round of selection and chose a set of encryption tools designed to be secure against attacks by future quantum computers. The four selected cryptographic algorithms are regarded as an important milestone in securing sensitive data against the possibility of cyberattacks from a quantum computer in the future [79].

The algorithms are created for the two primary purposes for which encryption is commonly employed: general encryption, which is used to secure data transferred over a public network, and digital signatures, which are used to verify an individual's identity. Experts from several institutions and nations collaborated to develop all four algorithms which are presented below.

- CRYSTALS-Kyber
  This cryptographic scheme is selected by NIST for general encryption and is based on the module Learning with Errors problem. CRYSTALS-Kyber is similar to the Ring-LWE cryptographic scheme but it is considered to be more secure and flexible. The parties that communicate can use small encrypted keys and exchange them easily with high speed.
- CRYSTALS-Dilithium
  This algorithm is recommended for digital signatures and relies its security on the difficulty of lattice problems over module lattices. Like other digital signature schemes, the Dilithium signature scheme allows a sender to sign a message with their private key, and a recipient uses the sender's public key to verify the signature but Dilithium

has the minor public key and signature size of any lattice-based signature scheme that only uses uniform sampling.

- FALCON
  FALCON is a cryptographic protocol which is proposed for digital signatures. The FALCON cryptosystem is based on the theoretical framework of Gentry et al [80]. It is a promising post-quantum algorithm as it provides capabilities for quick signature generation and verification. The FALCON cryptographic algorithm has strong advantages such as security, compactness, speed, scalability, and RAM Economy.
- SPHINCS+
  SPHINCS plus is the third digital signature algorithm that was selected by NIST. SPHINCS + uses hash functions and is considered to be a bit larger and slower than FALCON and Dilithium. It is regarded as an improvement of the SPHINCS signature scheme, which was presented in 2015, as it reduces the size of the signature. One of the key points of interest of SPHINCS+ over other signature schemes is its resistance to quantum attacks by depending on the hardness of a one-way function.

## 10. Conclusions

In recent years, significant progress has been made, taking us beyond classical computing and into a new era of data called quantum computing. Quantum research over the past few years has been particularly transformative, with scientific breakthroughs that will allow exponential increases in computing speed and precision. Research on post-quantum algorithms is active and huge sums of money are being invested for this reason, because it is necessary for the existence of strong cryptosystems.

It is considered almost certain that both the symmetric key algorithm and hash functions will continue to be used as tools of post-quantum cryptography. A variety of cryptographic schemes have been proposed for the quantum era of computing and this is a topic of ongoing research. The development and the standardization of an efficient post-quantum algorithm is the challenge of the academic community. What was once considered a science fiction fantasy is now a technological reality. The quantum age is coming and it will bring enormous changes; therefore, we have to be prepared.

**Author Contributions:** Investigation, G.C.M.; Writing—original draft, M.E.S.; Supervision, I.K.S., D.P. and G.G. All authors have read and agreed to the published version of the manuscript.

**Funding:** This research received no external funding.

**Conflicts of Interest:** The authors declare no conflict of interest.

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
