# Peer review of "Evaluation and Comparison of Lattice-Based Cryptosystems for a Secure Quantum Computing Era"

_electronics, doi:10.3390/electronics12122643_

Round 1
Reviewer 1 Report
This paper concerns abour quantum criptography. The rapid development of quantum computing devices woulb bring the possibility to tsolve a wide range of problems that traditional computers cannot. Therefore, quantum computers pose an enormous threat to encryption. Lattice-based cryptography is considered able to resist to a quantum computer attack and they are considered the future of post-quantum cryptography. The Authors list the most well-known lattice-based cryptosystems and present a systematic evaluation and comparison.
The paper is correct, well written and very interesting. for this reason I recommend the paper for publication.
Author Response
Thank you very much for your comments.
Reviewer 2 Report
Evaluation and Comparison of Lattice-based Cryptosystems for a Secure Quantum Computing Era
Authors: Maria Sabani, et al
This review manuscript addresses the lattice-based cryptosystems that are developed for a quantum computer based on lattice-based protocols by identifying their strengths and weaknesses. It’s believed that lattice-based cryptography is one way to deliver quantum-resistant encryption and are currently important candidates for post-quantum cryptography, which relies on the concepts of mathematical lattices.
Overall the paper is well written and a nice contribution to the literature on Lattice-based cryptography and should be interesting for the community who are studying these but needs substantial improvement by providing a more complete exposure to the field. I am happy to recommend the paper to be published in Electronics provided that clarifications made satisfactorily.
Detailed comments on the manuscript are provided below.
SPECIFIC COMMENTS (MAJOR)
1. The introduction is well written but needs expanding to make it clearer to a wider audience. For example, non-markovianity is useful to enhance security, see eg Vasile et al "Continuous variable quantum key distribution in non-Markovian channels", Phys. Rev. A 83, 042321 (2011) and photonic band gap media are promising to obtain nonMarkovian behaviour, Sci Rep 12, 11646 (2022). https://doi.org/10.1038/s41598-022-15865-5,
https://doi.org/10.1016/j.physleta.2022.128022.
2. I think it might be great to think about recent works on how phase modulation of coherent states play in quantum communication channels like in the paper: DOI 10.1088/0031-8949/90/7/074027 https://journals.aps.org/pra/abstract/10.1103/PhysRevA.92.012317 and the use of probabilistic noiseless linear amplifiers both at the encoding stage https://doi.org/10.1364/JOSAB.36.002938 where the information is coded on phase shifts and at the decoding stage https://journals.aps.org/pra/abstract/10.1103/PhysRevA.93.062315.
Author Response
Point 1: The introduction is well written but needs expanding to make it clearer to a wider audience. For example, non-markovianity is useful to enhance security, see eg Vasile et al "Continuous variable quantum key distribution in non-Markovian channels", Phys. Rev. A 83, 042321 (2011) and photonic band gap media are promising to obtain nonMarkovian behaviour, Sci Rep 12, 11646 (2022). https://doi.org/10.1038/s41598-022-15865-5,
https://doi.org/10.1016/j.physleta.2022.128022.
.
Response 1: Thank you very much for making us aware of this aspect. In Section 2, The evolution of Quantum Cryptography, and in Subsection 2, Quantum Key Distribution (Lines 148-157), we have included useful and interesting works about the enhancement of the security in the quantum key distribution protocol.
Point 2: I think it might be great to think about recent works on how phase modulation of coherent states plays in quantum communication channels like in the paper: DOI 10.1088/0031-8949/90/7/074027. https://journals.aps.org/pra/abstract/10.1103/PhysRevA 92.012317 and the use of probabilistic noiseless linear amplifiers both at the encoding stage https://doi.org/10.1364/JOSAB.36.002938 where the information is coded on phase shifts and at the decoding stage https://journals.aps.org/pra/abstract/10.1103/PhysRevA.93.062315.
Response 2: Again, we agree with the reviewer. Thank you. It was very useful and beneficial to all the papers that have been proposed as these recent works gave us another aspect of view and much useful information. So, we have included the proposed papers in lines 157-170.
Round 2
Reviewer 2 Report
Dear Authors,
The new version of the manuscript addresses most of the worries and I am therefore recommending acceptance of the manuscript.
There's minor correction in the references, as you can see in the introduction starts from (68). I suggest to properly place all the references accordingly.
Author Response
Response to Reviewer 2 Comments
Point 1: Dear Authors,
The new version of the manuscript addresses most of the worries and I am therefore recommending acceptance of the manuscript.
There's minor correction in the references, as you can see in the introduction starts from (68). I suggest to properly place all the references accordingly.
.
Response 1: Thank you very much for making us aware of this aspect. We agree again with the reviewer and we have changed the references’ structure.
